# CUDM: A Combined UAV Detection Model Based on Video Abnormal Behavior

**DOI:** 10.3390/s22239469

**Published:** 2022-12-04

**Authors:** Hao Cai, Zhiguang Song, Jianlong Xu, Zhi Xiong, Yuanquan Xie

**Affiliations:** Department of Computer Science, Shantou University, Shantou 515041, China

**Keywords:** UAV, malicious UAVs, object detection, video abnormal behavior

## Abstract

The widespread use of unmanned aerial vehicles (UAVs) has brought many benefits, particularly for military and civil applications. For example, UAVs can be used in communication, ecological surveys, agriculture, and logistics to improve efficiency and reduce the required workforce. However, the malicious use of UAVs can significantly endanger public safety and pose many challenges to society. Therefore, detecting malicious UAVs is an important and urgent issue that needs to be addressed. In this study, a combined UAV detection model (CUDM) based on analyzing video abnormal behavior is proposed. CUDM uses abnormal behavior detection models to improve the traditional object detection process. The work of CUDM can be divided into two stages. In the first stage, our model cuts the video into images and uses the abnormal behavior detection model to remove a large number of useless images, improving the efficiency and real-time detection of suspicious targets. In the second stage, CUDM works to identify whether the suspicious target is a UAV or not. Besides, CUDM relies only on ordinary equipment such as surveillance cameras, avoiding the use of expensive equipment such as radars. A self-made UAV dataset was constructed to verify the reliability of CUDM. The results show that CUDM not only maintains the same accuracy as state-of-the-art object detection models but also reduces the workload by 32%. Moreover, it can detect malicious UAVs in real-time.

## 1. Introduction

Unmanned aerial vehicles (UAVs) have various shapes and sizes and can fly quickly at different altitudes [1,2]. These features allow UAVs to perform many tasks that humans cannot accomplish. Therefore, UAVs have been extensively utilized in different fields, such as communication [3], ecological survey [4], and agriculture [5]. Most scenarios where UAVs are used relate to the analysis of onboard captured images. This is because UAVs can fly to places that are inconvenient or inaccessible to humans for data collection. These data provide researchers with a wide variety of information. Researchers can use this information to conduct relevant research. For example, the data captured by UAVs can be used for traffic collision risk analysis [6], analysis of the agricultural environment [7] and for monitoring the state of crops [8], etc. However, UAVs are also employed by malicious users to, for example, illegally collect information, violating the privacy of other people [9]. Even more dangerous is the use of UAVs to carry explosives or other dangerous items for terrorist attacks [10]. Therefore, UAV detection is highly necessary for controlling these vehicles.

The conventional method of detecting objects uses physical information [11,12] for active or passive object detection. For example, monitoring devices can passively receive physical information. When a UAV is on a flight mission, it generates physical information, such as flight noise and electronic signals. This physical information can be captured using monitoring devices. The presence of a UAV can be determined by analyzing the physical information received by the monitoring devices. In addition, radars can actively use physical information to detect UAVs. A radar works by actively emitting electromagnetic waves, which are reflected back to the radar when electromagnetic waves contact an object. Then, by analyzing the reflected electromagnetic waves, it is possible to determine whether the object is a UAV or not.

With the development of artificial intelligence, deep learning-based methods have been developed for UAV detection. Several methods have also been used for analyzing images to detect UAVs [13,14,15,16], which typically require deep neural networks. However, a deep neural network must be trained before it can be used. The dataset used for training must contain UAV images. Through training, the neural network can autonomously determine whether a detected image contains a UAV or not. Specifically, neural networks learn the UAV characteristics through training. Neural networks then need to determine whether the detected image contains the characteristics of UAVs.

Although the above methods are feasible for detecting UAVs, there are still some problems. First, physical information, which can be limited by distance and the environment, is used. In the case of long distances or complex environments, the monitoring device may not capture flight noise or electronic signals. Consequently, these devices may not be able to detect UAVs on the scene. In addition, the physical information generated by non-UAVs may be similar to that generated by UAVs. Therefore, a single type of physical information is not sufficient and multiple pieces of physical information are necessary to identify whether an object is a UAV. However, collecting multiple pieces of physical information requires more monitoring devices, which increases the cost of detection. Second, although the methods employing deep neural networks can detect UAVs more easily than those using physical information, they also face the problem of a long detection time. For example, deep neural networks detect UAVs by comparing the features learned through training with the features of the object in the image. If the features match, the unknown object can be identified as a UAV. However, for a dataset with many images that do not contain a UAV, deep neural networks still need to perform feature comparisons for each image. Comparing features in images that do not contain UAVs is inefficient and only increases the detection time.

Overall, the development of deep learning has greatly helped in the task of UAV detection. The advantages of deep learning are clear. Firstly, deep learning requires less equipment, and only simple equipment is needed to complete the task of UAV detection. Secondly, deep learning makes it easier to fuse multiple pieces of information, which can help researchers to complete UAV detection tasks in complex conditions. Finally, deep learning can help researchers in terms of computational work, allowing them to focus on deeper research, thus advancing the work of UAV detection. While deep learning has a great number of advantages in the work of UAV detection, there are still some problems that require attention. For example, the time cost of deep learning models. In general, deep learning is used to accomplish object detection by learning target features. However, different deep learning models spend different amounts of time learning target features. Therefore, a suitable deep-learning-related model is very important for UAV detection work, otherwise it may lead to significant time costs.

In this study, we propose a detection model called the combined UAV detection model (CUDM), which is based on analyzing abnormal video behavior. The video is cut into images at certain time intervals for detection. Then, the conventional object detection process is split into two steps. The first step is to determine the presence of an object and the second step is to identify the object type. We added abnormal behavior detection to the first step to screen images that do not contain UAVs. With this improvement, we can solve the problem of a long detection time.

The remainder of this paper is organized as follows. Section 2 presents some studies on UAV detection and abnormal behavior detection. Section 3 describes the structure of the proposed framework and details of CUDM. Section 4 discusses the results of the experiments on a self-made dataset and comparison with some baseline models. Section 5 concludes the study and suggests future work.

## 2. Related Work

Most studies on UAV detection utilize deep learning because it has shown adaptability in the field of object detection and achieved satisfactory results [17,18,19]. Deep learning is continuously developing, providing more references for deep learning-based research.

### 2.1. Deep Learning-Based Models

Xu et al. used unsupervised training and designed a multilayer autoencoder for learning appearance and features [20]. Their work has expanded the methods that can be used for training. Bochkovskiy et al. proposed the YOLOv4 model, which has a faster detection speed and higher detection accuracy [21]. YOLOv4 has also been used as a baseline model in many studies. In addition, Tan et al. proposed the EfficientDet model [22], which extends the classical feature pyramid model and adopts the strengths of other models. EfficientDet has attracted much attention from researchers, and more people have conducted relevant outreach research. All of these studies on deep learning serve as reference materials for UAV detection.

### 2.2. Deep Learning in UAV Detection

Many studies on UAV detection are based on existing deep learning research. Seidaliyeva et al. attempted to detect UAVs using YOLOv2 [13], which is a well-known deep learning-based method. Kassab conducted experiments on UAV target tracking using deep learning and achieved success in this area following [14]. Zhao et al. proposed the UAVDet network for detecting low-flying UAVs, which still uses deep learning as a baseline [1]. Although all of these studies have accomplished UAV detection using deep learning, they have one common characteristic, i.e., they only make use of static and unrelated image information. In these studies, the dataset only needs to contain the target, and it is not necessary to consider the links between data [13,19,22]. This characteristic can be considered an advantage in deep learning because the limitations of the dataset are lower. However, it is not suitable for detecting abnormalities.

### 2.3. Abnormality Detection

Abnormality detection is commonly used in detection studies where behavioral differences are evident, and most of these studies are based on videos taken with a fixed camera. This means that the data in abnormality detection are dynamic and interrelated. Considering that the scene exhibits an evident distinction when a UAV is present, using abnormality detection for UAV identification is a novel approach. There has been research using abnormal detection to identify UAVs [23], but it is still relatively scarce. However, there is extensive research on abnormal detection and these results can be used as an important reference.

Many studies have been conducted for abnormality detection, which refers to the identification of events that do not conform to expectations [24]. The goal of abnormality detection is to identify anomalies. Normal feature reconstruction [25] is a common strategy in this field. In Refs. [26,27], a manual feature-based approach for reconstruction, which learns dictionaries to reconstruct normal events, is used. The approach has a large error, so there is no guarantee in terms of the results of the reconstruction. In Refs. [18,28,29], deep learning-based methods were used for reconstruction. However, deep learning-based methods contain very deep networks, meaning that accuracy is not ensured when reconstructing abnormal events. The above methods determine anomalies by reconstructing the training data. Another approach is using video frame prediction to determine anomalies [25]. In addition, other abnormality detection studies can be used as important references. Jia proposed a framework for the pattern-based discovery of abnormal behaviors [30]. The framework defines criteria for patterns and proposed a mining algorithm for abnormal behavior. Sun et al. proposed a weakly supervised abnormality detection method based on temporal consistency [31]. This method can infer the start and end frames by learning the relevant features, which allows this process to become automatic. However, inference errors cause significant degradation in the effectiveness. Iwashita et al. investigated the use of privacy-preserving videos to detect abnormal behaviors [32]. This method achieved better detection results; however, the dataset had several limitations. Yuan et al. proposed an identification method to quickly detect the abnormal behavior of pedestrians [33]. In their study, the backbone features of the target were extracted to remove other features to improve target accuracy. Ito et al. used the OS-ELM and an autoencoder for adaptive abnormal behavior detection [34]. This method has high accuracy in detecting known abnormal behaviors; however, unknown anomalies cannot be detected.

## 3. Methodology

### 3.1. Framework of CUDM

The framework and workflow of CUDM are shown in Figure 1. The workflow is divided into two stages. The first stage is to detect abnormal behavior in the video, which involves detecting abnormal objects in the sky. The second stage is to identify anomalies in the images with abnormal behavior from the first stage and to determine whether UAVs exist. The first stage is composed of the following three main modules: (1) module for generating future prediction frames; (2) module for checking the quality of the generated prediction frames; and (3) module for distinguishing and detecting anomalies. The main component of the second stage is the YOLOv4-based anomaly recognition module. Let $I_{t-n}$∼$I_{t-1}$ denote the past frames, $I_{t}$ the actual future frames, and ${\hat{I}}_{t}$ the predicted future frames. The working process of CUDM is as follows: First, $I_{t-n}$∼$I_{t-1}$ are used to generate ${\hat{I}}_{t}$. The quality of the generated prediction frame module is verified by evaluating the quality of ${\hat{I}}_{t}$ using intensity loss, gradient loss, and optical flow loss and providing feedback to the module for the generation of future prediction frames, which is used to help in the self-learning of the generating module. Second, the peak signal-to-noise ratio (PSNR) is used to distinguish and detect anomalies to determine whether an object with abnormal behavior appears in $I_{t}$ using the difference in the PSNRs between $I_{t}$ and ${\hat{I}}_{t}$. Finally, $I_{t}$ with abnormally behaving objects is input to the YOLOv4-based anomaly recognition module to determine the specific class of the objects.

### 3.2. Generating Future Prediction Frames

In the CUDM framework, future frames are predicted based on past frames. However, if the predicted future frames are incorrect and unclear, errors and even misjudgments will occur when comparing the predicted with the actual future frames. Frame problems occur due to the structural asymmetry of the frame generation network, causing gradient disappearance and information imbalance [18,35]. To solve these problems, a modified U-Net network [25], as illustrated in Figure 2, is used.

We studied other works that used U-Net networks and found that the majority of studies used a four-layer structured U-Net; therefore, we decided to follow this design to maximize the performance of our modified U-Net network. In addition, we made some improvements to make it easier for the U-Net model to achieve information symmetry. Specifically, we kept the output resolution unchanged between each of the two convolutional layers. In this way, the U-Net network no longer requires cropping and resizing when combining information.

As depicted in Figure 2, when the past frames enter the modified U-Net network, the following processes occur:(1)Feature extraction from past frames. First, one convolution is performed in the same layer of the submodule, and the size of the convolution kernel is 3 × 3. The resolution of the frame remains the same after each convolution, but the number of channels changes, which in turn changes the frame specification from (288, 512, 3) to (288, 512, 64). Second, max pooling is performed using a kernel size of 2 × 2. After max pooling, the resolution becomes half of that of the previous submodule, but the number of feature maps remains the same as that of the previous layer of submodules. The image specification changes from (288, 512, 64) to (144, 256, 64). Finally, the above steps are repeated, and the frame specification becomes (36, 64, 512). At this point, the modified U-Net network completes the extraction of features from different layers.(2)Image recovery by each layer feature. First, up-convolution is performed, and the size of the convolution kernel is 3 × 3. The frame changes from (36, 64, 512) to (72, 128, 512). However, the results of the up-convolution and max pooling with the same resolution are fused to obtain the final recovered picture. Second, deconvolution is performed. The size of the convolution kernel is 3 × 3, similar to that of up-convolution, and the frame is changed from (72, 128, 512) to (72, 128, 256). Finally, the above steps are repeated layer by layer to recover and generate the picture.

### 3.3. Checking the Quality of the Generated Prediction Frames

Generating predicted future frames using the modified U-Net network relies on the extraction and fusion of different levels of the frame features. However, owing to the different resolutions or number of channels of each layer of features, the generated frames obtained by decoding the features may encounter problems, such as chromatic aberration and pixelation. In addition, the problem of inaccurate object orientation may occur because moving objects are irregular, and the modified U-Net only restores moving objects based on picture features.

#### 3.3.1. Chromatic Aberration Problem

To solve the chromatic aberration problem, this study adopts the intensity constraint approach to ensure that each generated pixel in the RGB three primary color space resembles a real pixel. Let $\hat{I}$ and *I* denote the intensity vectors of the predicted and actual future frames, respectively, both of which are derived from the set of frames under operation frames. Additionally, let α be the set intensity error threshold, which is obtained from the training process of the model. Therefore, the minimization of the intensity constraint can be represented by Equation (Equation 1).
(1)$$L_{is}\left(I,\hat{I}\right)={\min_{I,\hat{I}\in frames}∥\hat{I}-I∥}^{2}$$
where $L_{is}$ is the absolute minimum value of the difference between the intensity vector of the predicted and actual future frames. When $L_{is}\leq \;\alpha$, the generated frame is considered eligible for this intensity constraint, which means that the predicted future frame is usable.

#### 3.3.2. Pixelation Problem

For the pixelation problem of the generated frame, this study uses the gradient constraint approach [35] to make the generated frame clearer. Let ${\hat{I}}_{t}$, ${\hat{I}}_{t-1}$, $I_{t}$, and $I_{t-1}$ represent the gradient vectors of the predicted and actual future frames at time *t* and t−1, respectively, and β the set gradient error threshold, which is obtained from the training process of the model. Consequently, the gradient loss can be expressed by Equation (Equation 2).
(2)$$L_{gd}\left(I,\hat{I}\right)=∥\left|{\hat{I}}_{t}-{\hat{I}}_{t-1}\right|-|I_{t}-I_{t-1}|∥$$
where $L_{gd}$ denotes the absolute minimum value of the gradient vector difference between the predicted and actual future frames. When $L_{gd}\leq \;\beta$, the gradient difference fulfils the demand, which implies that the frame satisfies the gradient constraint and can be used.

#### 3.3.3. Moving Object Problem

To solve the problem of moving objects, this study uses FlowNet [36,37] to calculate the optical flow constraints. Let *f* denote FlowNet; ${\hat{I}}_{t}$, $I_{t}$, and $I_{t-1}$ represent the predicted and actual future frames at time *t* and t−1, respectively; and γ, which is obtained from the training process of the model, denote the set of optical flow error thresholds. Thus, the mathematical description of the optical flow constraints is given by Equation (Equation 3).
(3)$$L_{of}\left(I_{t-1},I_{t},{\hat{I}}_{t}\right)=∥∥f({\hat{I}}_{t},I_{t-1})∥-∥f(I_{t},I_{t-1})∥∥$$
where $L_{of}$ denotes the absolute value of the minimum difference between the optical flow estimation of the predicted and actual future frames. When $L_{of}\leq \;\gamma$, the requirement is satisfied. This study uses the predicted future frame, actual future frame, and past frames for optical flow estimation. Because the presence of subtle differences in the frames can also lead to widely different optical flow maps, ${\hat{I}}_{t}$, $I_{t}$, and $I_{t-1}$ are used as inputs in FlowNet. FlowNet converts the frames into optical flow maps, and then compares the optical flow maps of the predicted and actual future frames at time *t* with the actual future frame at time t−1, which returns their optical flow values. Then, the difference between them is calculated to obtain the optical flow difference. When the optical flow difference is limited to within γ, the generated picture results can be considered usable.

### 3.4. Distinguishing and Detecting Anomalies

To distinguish the frames containing abnormal behavior from normal frames, this study adopts the idea of judging the future based on the normal past. Let *F* denote the function that determines whether a frame is abnormal, *I* represent all frames, and $I_{abnormal}$ indicate the frames that are judged as abnormal by *F*. Thus, the frames with abnormal and normal behaviors can be distinguished using Equation (Equation 4).
(4)$$F{\left(I\right)}_{I\in allframes}=I_{abnormal}$$

Anomalies are difficult to identify. Whether some behaviors are abnormal or not needs to be analyzed in the scenes. For example, the goal of this study is to detect the existence of UAVs. Thus, the existence of a UAV in a scene is defined as abnormal and its absence in a scene is considered normal. CUDM can determine normal and abnormal behaviors based on the number of scenes. Most scenes do not usually contain UAVs, and only a very small number of scenes have UAVs. Therefore, most scenes without UAVs are considered normal, and the very few scenes with UAVs are considered abnormal.

CUDM finds anomalies by comparing the generated predicted future frame ${\hat{I}}_{t}$ with the actual future frame $I_{t}$. Specifically, it first generates ${\hat{I}}_{t}$ based on the past normal frame. As there is no UAV in the past normal frame, ${\hat{I}}_{t}$ will not contain a UAV. However, when $I_{t}$ contains a UAV, there is a clear difference between $I_{t}$ and ${\hat{I}}_{t}$. To quantify this difference, the PSNR is used in CUDM. Assuming that Δ denotes the PSNR difference between $I_{t}$ and ${\hat{I}}_{t}$, Equation (Equation 5) is the mathematical expression for this quantification.
(5)$$\Delta_{\left({\hat{I}}_{t},I_{t}\right)}=|PSNR_{{\hat{I}}_{t}}-PSNR_{I_{t}}|$$

CUDM evaluates each frame and calculates its PSNR. A high PSNR for the *t*-th frame indicates that it is likely normal. During training, all frames learned by CUDM are normal frames. The PSNR decreases when the detected frame contains a UAV. Thus, CUDM can determine which frame contains UAVs based on the PSNR change curve.

### 3.5. YOLOv4-Based Anomaly Recognition

CUDM uses an abnormal identification method based on YOLOv4 [21] to determine whether a suspicious flying object is a UAV. In selecting a model for CUDM to identify abnormal objects, we took note of the YOLO family. After considering stability, adaptability, and reference, we chose YOLOv4. In addition, we determined the parameters used in our study by examining a large number of works using YOLOv4. Figure 3 displays the structure and process of YOLOv4.

As shown in Figure 3, the backbone feature extraction network of YOLOv4 consists of the following four parts: CSPDarknet53, SPP, PANet, and YOLO head. CSPDarknet53 is involved in image feature extraction. When an image containing a UAV is input into CSPDarknet53, first, the model resizes the image to a resolution of 480 × 480, and the number of channels is 3, which is denoted as (480, 480, 3). Second, the image after resizing is convolved to obtain a feature layer with a resolution of 480 × 480 and 32 channels, denoted as (480, 480, 32). Finally, the feature layer obtained in the previous step is downsampled to obtain a feature layer with a resolution of 240 × 240, and the number of channels is 64, which is denoted as (240, 240, 64). Downsampling is continued for this feature layer to obtain different levels of feature layers with specifications of (120, 120, 128), (60, 60, 256), (30, 30, 512), and (15, 15, 1024). They are then fed into a feature pyramid consisting of SPP and PANet.

The feature pyramid performs a cross-fusion of these features to achieve enhanced feature extraction. Specifically, in the SPP, the model enhances the knowledge of UAV features. In addition, the model separates the most significant UAV contextual features. In the PANet, the semantic features of the UAV are analyzed from top to bottom. Simultaneously, the location features of the UAV are analyzed from bottom to top. The feature information is aggregated from different levels of feature layers for convolution and upsampling, because it can improve the feature extraction performance and model ability to detect UAVs. Finally, three UAV enhancement features are obtained from the feature pyramid: (60, 60, 128), (30, 30, 256), and (15, 15, 512). These enhanced features are fed into the YOLO head.

The YOLO head performs decoding. First, 3 × 3 and 1 × 1 convolution operations are used for feature integration and channel number adjustment. Consequently, the shapes of the three enhanced feature layers are adjusted to (15, 15, 27), (30, 30, 27), and (60, 60, 27). Specifically, N images of the UAV with a resolution size of 480 × 480 are input, and the three shapes are finally obtained as (N, 15, 15, 27), (N, 30, 30, 27), and (N, 60, 60, 27). Second, for detecting objects of different sizes, the YOLO head divides the image into 15 × 15, 30 × 30, and 60 × 60 grids. These grids represent the position of the prediction box in the image. The prediction box is used to determine whether the box contains a UAV. A confidence level of between 0 and 1 is used for this determination. This confidence level represents the probability that the model can identify the object in the box as a UAV. Finally, screening is performed to exclude targets with confidence levels lower than a predefined threshold, and the prediction box with the highest confidence level is selected as the final result.

Algorithm 1 is the main algorithm of CUDM after combining the anomaly identification stage. In this algorithm, $\Delta_{limit}$ is used to determine whether the PSNR value is abnormal; FetureMap represents the feature layer obtained after convolution; and MaxFetureMap indicates the enhanced feature after cross-fusion. In the process, steps 1 to 17 are the necessary steps for detecting abnormal behavior in the video, whereas steps 18 to 22 are used for identifying anomalies. The algorithm first requires inputs of past frames and the determination of thresholds for intensity, gradient, optical flow, and PSNR. Subsequently, the prediction frame is generated, and each loss is calculated. If the loss meets the condition, CUDM will perform an abnormality judgment, and if the frame is abnormal, CUDM will perform abnormal object recognition.
**Algorithm 1** Main algorithm of CUDM**Input:** 
Past Frames $I_{t-n}∼I_{t-1}$, Actual Future Frame $I_{t}$, Intensity Error Threshold α, Gradient Error Threshold β, Optical Flow Error Threshold γ, PSNR limit Δ**Output:** 
UAV Detection Result $I_{final}$1:**for** $I\in I_{t-n}∼I_{t-1}$**do**2:   ${\hat{I}}_{t}=$U-Net$\left(I_{t-4},I_{t-3},I_{t-2},I_{t-1}\right)$3:   **while** 
${\hat{I}}_{t}$
**do**4:    $L_{is}\left(I_{t},{\hat{I}}_{t}\right)={\min_{I_{t},{\hat{I}}_{t}\in frames}∥{\hat{I}}_{t}-I_{t}∥}^{2}$5:    $L_{gd}\left(I_{t},{\hat{I}}_{t}\right)=∥\left|{\hat{I}}_{t}-{\hat{I}}_{t-1}\right|-|I_{t}-I_{t-1}|∥$6:    $L_{of}\left(I_{t-1},I_{t},{\hat{I}}_{t}\right)=∥∥f({\hat{I}}_{t},I_{t-1})∥-∥f(I_{t},I_{t-1})∥∥$7:    **if** $L_{is}\leq \alpha$ and $L_{gd}\leq \beta$ and $L_{of}\leq \gamma$
**then**8:      $PSNR\left({\hat{I}}_{t},I_{t}\right)=\left|PSNR_{{\hat{I}}_{t}}-PSNR_{I_{t}}\right|$9:      **if** $PSNR\left({\hat{I}}_{t},I_{t}\right)\leq \Delta_{limit}$ **then**10:       $I_{abnormal}=I_{t}$11:       go to step 18.12:      **end if**13:     **else**14:      go back to step 2.15:     **end if**16:    **end while**17:**end for**18:Resize $I_{t}$ become resolution of 480 × 480 $\to {I_{t}}^{\prime }$19:Convolute ${I_{t}}^{\prime }\to FetureMap$20:Cross fusion FetureMap→MaxFetureMap21:Decode $MaxFetureMap\to I_{final}$22:**return** $I_{final}$


## 4. Experiments

In this study, experiments were conducted on a self-made civilian UAV dataset to evaluate the CUDM performance. This study attempts to address the following questions.

RQ1. How well does CUDM detect UAVs based on abnormal behavior detection?RQ2. How do different parameter settings affect the performance of CUDM?

### 4.1. Dataset

We used fixed cameras and small civilian UAVs to produce datasets. We obtained the dataset using a camera to capture the UAV. The camera faced toward the sky with a fixed angle. The UAV appearances ranged from low to high altitudes. Low-altitude images contain a small number of buildings, whereas high-altitude images contain pure sky. The data were processed differently during the two stages of detection of abnormal behavior and the identification of abnormal objects. This is because the training was performed in an unsupervised manner for abnormality detection. However, a supervised method was used for training in the target detection.

#### 4.1.1. Detection of Abnormal Behavior Stage

In this stage, a dataset was obtained from the video of a single UAV flying in different backgrounds. It contains the following three types of images: UAV flying at low altitudes, UAV flying at high altitudes, and no UAV flying in the sky. The dataset contains 1800 training images and 2200 test images with a resolution of 512 × 288 pixels. In the training images, there are 1400 low-altitude images and 400 high-altitude images, none of which contain UAVs (they are normal). In the test images, there are 1600 low-altitude images and 600 high-altitude images, with 1533 images containing UAVs (they are abnormal). Figure 4 illustrates a part of the dataset for detecting abnormal behavior. This dataset does not require labeling of UAVs in the images and is trained using an unsupervised method.

#### 4.1.2. Anomaly Identification Stage

In this stage, a dataset was obtained from the video of a single UAV flying in different backgrounds. It contains two types of images: UAV flying at low altitudes and UAV flying at high altitudes. All images contain UAVs. The dataset comprises 3200 training images and 400 test images with a resolution of 480 × 480 pixels. In the training images, there are 600 low-altitude images and 2600 high-altitude images, all of which contain UAVs. In the test images, there are 200 low-altitude and 200 high-altitude images. The dataset includes most of the possible attitudes, distances, and bearings of UAVs in flight. This allows the model to learn the characteristics of UAVs better. A part of the dataset from this stage is depicted in Figure 5. This dataset require labeling of UAVs in the images.

### 4.2. Evaluation Metrics

#### 4.2.1. Indicators Used in the Stage of Detecting Abnormal Behavior in the Video

The mean-square error (MSE) [38] has been commonly used to assess the quality of generated frames by calculating the Euclidean distance between each pixel in the RGB color space of the predicted and actual future frames. However, it has been demonstrated that PSNR is a better method for evaluating generated frame quality [35]. Therefore, the PSNR is used in this study. The MSE needs to be known first to calculate the PSNR. Let *m* × *n* denote the resolution of the frame, $I_{t}$ the actual future frame, ${\hat{I}}_{t}$ the predicted future frame, and *j*, *k* the subscripts in a two-dimensional matrix. Thus, the MSE can be expressed as shown in Equation (Equation 6).
(6)$$MSE\left(I,\hat{I}\right)=\frac{1}{mn}\sum_{j=0}^{m-1}\sum_{k=0}^{n-1}{\left[I_{t}\left(j,k\right)-\hat{I_{t}}\left(j,k\right)\right]}^{2}$$

The PSNR is calculated after obtaining the MSE. Let $MAX_{I}$ represent the maximum value of the image point color. The PSNR is calculated using Equation (Equation 7). For $MAX_{I}$, if each point color is sampled with eight bits, then $MAX_{I}$ is 255.
(7)$$PSNR\left(I,\hat{I}\right)=10\log_{10}\frac{{\left[MAX_{I}\right]}^{2}}{MSE}$$

A higher PSNR is due to a low MSE, because a small error between the actual and predicted future frames results in a low MSE. However, a predicted future frame generated from a normal frame must differ from an actual future frame with an anomaly, which increases the MSE and decreases the PSNR. This phenomenon can be used to determine the existence of anomalies.

#### 4.2.2. Indicators Used in the Anomaly Identification Stage

To evaluate the detection effectiveness of CUDM, we employed other models to process the same dataset. These different models were used for cross-sectional comparisons with CUDM and experiments were performed to test the effectiveness of the dataset constructed in this study. The following metrics were used for the evaluation: recall, precision, F1, AP, mAP, and amount of data.

(1)Recall indicates the number of actual samples that are correctly predicted.(2)Precision denotes the number of samples correctly predicted by the model. Here, TP indicates a positive prediction that is actually positive, FP denotes a positive prediction that is actually negative, FN represents a negative prediction that is actually positive, and TN indicates a negative prediction that is actually negative. Recall and Precision are calculated using Equations (8) and (9), respectively.
(8)$$Recall=\frac{TP}{TP+FN}\times 100\%$$
(9)$$Precision=\frac{TP}{TP+FP}\times 100\%$$

The higher the values of the two evaluation metrics, the better the performance. However, in extreme cases, the two values can be polarized. Therefore, F1 was used to evaluate the balance between Recall and Precision. The formula for calculating F1 is shown in Equation (Equation 10).
(10)$$F1=\frac{2}{\frac{1}{Precision}+\frac{1}{Recall}}=\frac{2\times PrecisionRecall}{Precision+Recall}$$

(3)F1 can be considered as a reconciled average of the model accuracy and recall. A higher F1 score indicates a better balance between the accuracy and recall, which denotes a better model performance.(4)AP indicates the average of the highest Precision at different Recall values.(5)mAP denotes the average of the AP values for all categories.

AP and mAP were used to evaluate the accuracy of the model. The higher the value, the better the accuracy.

In addition, this study compares the amount of data that must be detected when determining UAVs using different models to assess the advantage of CUDM.

### 4.3. Performance Comparison (RQ1)

To evaluate the effectiveness of CUDM, this study compares it with the following models.

YOLOv3 [39]. YOLOv3 is the third version of the YOLO (You Only Look Once) family of target detection algorithms. It was selected as one of the comparison models in this study because YOLOv4 and YOLOv3 are similar. UAVs typically appear as small targets, and v4 improved the detection of small targets. Therefore, the comparison can show the advantages of v4 over v3 in UAV detection.

Single-shot multibox detector (SSD) [40]. The SSD is a one-stage, multi-box prediction method for object detection. It uses a convolutional neural network (CNN) for direct detection. The SSD has two features in the detection of small objects. First, it adopts different scales of feature maps for detection. More forward-located feature layers are easier to obtain. Therefore, more forward-located feature maps are used to detect small objects, which improves the detection performance. Second, anchors with different scales and aspect ratios are used to improve the performance of detecting small objects. The SSD was chosen as one of the references to evaluate CUDM for UAV (small target) detection.

Faster R-CNN [41]. Faster R-CNN is a very effective object detection algorithm. It is also the basis of many object detection algorithms. As a two-stage algorithm, it is more complex and the detection accuracy is higher. Faster R-CNN first obtains the candidate frames through a CNN and then performs classification and regression. Such step-by-step optimization improves the detection accuracy of the model. Faster R-CNN was used as a reference model for CUDM to evaluate its performance in terms of detection accuracy.

EfficientDet [22]. EfficientDet is an object detection model proposed by the Google Brain team, which achieved new SOTA results at that time. The model was designed with reference to other good neural networks. First, it uses residual neural networks to increase the network depth and obtain features in deeper layers. Second, the number of layers of features extracted from each layer is changed to increase the width of the network. Finally, it increases the resolution of the input image to enable the network to learn more effectively, which helps to improve the accuracy. A comparison of CUDM with EfficientDet can effectively evaluate the performance of CUDM in object detection.

As mentioned above, these models were chosen as comparative models based on a combination of considerations. Similarly, we set the parameters for these comparative models based on a wide range of information and experience. Firstly, we have read a large number of studies that use these models. We summarised the setting of the model parameters in these studies to obtain a range of parameters for the models to perform well. Secondly, we experimented with each of the data within the parameter ranges. The final experimental parameters were chosen based on a combination of considerations.

We used four UAVs with distinct shapes, named Category1, Category2, Category3, and Category4, as experimental targets. The main results are presented in Table 1 and Table 2. From the results, it can be seen that CUDM performs better in the dataset built in this study. Some of the results are even better than those of the existing models. The following is an analysis of the experimental results.

First, the experiments show that the most prominent advantage of CUDM is the reduction in data volume. The test dataset size is 2.2k, and because some of the images in the test dataset do not include UAVs, CUDM only needs to detect 1.5k of them. This results in a 32% work reduction. This value increases as the number of times the UAV appears in the frame decreases. This is because, first, the other models detect objects by processing each video frame. Second, the other models determine if the frame contains a UAV by identifying whether all objects contain features of the UAV. However, CUDM detects abnormalities first before deciding whether to perform UAV recognition on a frame. In addition, looking at the average Recall, average Precision, average F1 and average AP for each model, some of the CUDM metrics are the highest metrics and others are about the same as the highest metrics. It can be said that the overall performance of CUDM is in compliance with the requirements.

Second, from the comparison of the mAP values, it can be seen that CUDM performs slightly better than YOLOv3. This proves that CUDM based on YOLOv4 is better than YOLOv3 for UAV detection. This also indicates that YOLOv4 has no negative improvement in small object detection. SSD is good for small object detection. The mAP of CUDM is close to that of SSD, which means that CUDM performs sufficiently well for detecting small objects such as UAVs. However, considering its flexibility, CUDM should not be used only for detecting small targets such as UAVs. Therefore, we did not use SSD as the base framework for the anomaly identification stage. The mAP values of Faster R-CNN and EfficientDet are lower than that of CUDM. This is mainly due to our hardware limitations. These two models, which have theoretical advantages, do not perform as expected, showing that they are not suitable for UAV detection under the conditions of pursuing low cost.

Finally, in terms of the model recall for each category, CUDM performs well and consistently. The other models have significantly lower recall values for a particular category. In addition, we compared the precision. For CUDM, there is a large difference between precision and recall in the detection of Category1. However, CUDM maintains a balance between precision and recall for the detection of the other categories. Thus, the F1 value of CUDM is higher than that of the other models in most of the detections. This means that the performance of CUDM is more balanced.

In addition to comparisons with standard well-known deep networks, we have also compared it with other systems that solve the UAV detection problem. The results of the comparisons are shown in Table 3.

We used the following six evaluation criteria to compare these models: Data set acquisition, Data type, Variety of experimental scenarios, Low experimental costs, Real-time detection, and UAV detection results. Where N represents “No” and Y indicates “Yes”. The results show that CUDM performs better than any other system in its category. The following is an analysis of the experimental results.

Firstly, after studying a large number of UAV detection works, it was found that there are no other suitable datasets published in the field of UAV detection. So, all the datasets used in the models in Table 3 are our own. In addition, each UAV detection system uses a different type of data. Therefore, it can be considered that CUDM is compared with a diverse range of UAV detection systems, thus demonstrating the effectiveness of CUDM.

Secondly, we evaluate the comprehensiveness of the CUDM using the following four metrics: variety of experimental scenarios, low experimental costs, real-time detection and UAV detection results. These four metrics look at whether the model can work in complex scenarios, whether the model is low-cost, whether the model is real-time and whether the model is able to detect drones. From the results, CUDM can meet these four indicators, while the other models have certain shortcomings.

### 4.4. Influencing Factors of CUDM (RQ2)

The most important difference between CUDM and conventional object detection models is the inclusion of the abnormal behavior detection stage. Some settings in this stage influence the performance of CUDM. We mainly investigated the following influencing factors: continuity of screen frames and training duration.

#### 4.4.1. Effect of Continuous and Discontinuous Frames

Continuous and discontinuous frames produce different generative frames, which may affect the model training. Therefore, we performed experiments using controlled variables. When intercepting frames from a video, the smaller the spacing of frames, the stronger the continuity. We intercepted two sets of frames, namely continuous and discontinuous frames, with different frame spacings in the same video. We used these two sets for frame generation and training. The results of frame generation (Figure 6) show that discontinuous frames lead to serious and blurred residual images, which are not conducive to the calculation of the PSNR. As illustrated in Figure 7, the PSNR varies during the training process. It can be observed that the PSNR values of the discontinuous group are generally small, reaching only a maximum of approximately 30. However, the PSNR values of the continuous group usually exceed 40 [47]. As a comparison, the PSNR of the continuous group reaches a peak of 40+ for the same settings. This indicates that frame coherence is highly important.

#### 4.4.2. Effect of Training Duration

The training duration has an impact on the effectiveness of model training. The higher the number of steps, the longer the training duration. In general, the effectiveness does not grow indefinitely with an increase in training duration. Thus, we set different training durations to explore the optimum value in which CUDM can achieve cost-effective results. As displayed in Figure 8, comparing the results of the 3000 and 5000 steps, we can see that the model is very close to the maximum limit after 3000 steps. This indicates that extending the training time after 3000 steps is meaningless, as it will only increase the cost. Second, comparing the 1000 steps with the other two steps, it is evident that at 1000 steps, the PSNR is still far from the peak value, and we cannot obtain an effective PSNR. In addition, when the training duration is extremely low, anomalies cannot be detected by the PSNR change curve in some cases with little differentiation. Therefore, the training duration of 3000 steps is the best choice for this experiment.

### 4.5. Demonstration of CUDM Effectiveness

We reduced the frame space to make continuous frames and set the training duration to a suitable length. As shown in Figure 9, the curve changed significantly, indicating that the UAV appears in the surveillance footage. CUDM uses this change to determine whether a UAV is present. Owing to the use of continuous frames, the PSNR is generally high, with maximum values reaching 40+ and minimum values of greater than 30. CUDM creates a value space for the difference between abnormal and normal values. Thus, it can find the abnormality by the change in the PSNR value.

## 5. Conclusions

In this study, we propose a detection model called CUDM, which is based on analyzing video abnormal behavior. We primarily used video surveillance footage and abnormal behavior detection. Video surveillance footage is the source of data, and we used an abnormal behavior detection model to filter out useless images. A large number of images can be generated with video surveillance. CUDM improves the conventional object detection process by solving the problem of processing many useless images. To achieve this, abnormal behavior detection is performed before object detection to remove images that do not contain abnormal objects (normal frames), which account for most of the data. Subsequently, a small number of screened images containing abnormal objects (abnormal frames) are identified at the recognition stage for determining the type of abnormal objects. The experiments showed that detecting abnormalities significantly reduces the work of the recognition stage by 32%. CUDM still maintains the same level of accuracy as the existing detection methods but the conventional methods require a full set of recognition processes for each normal image. CUDM only needs to compare the PSNR between frames when processing normal images. Thus, CUDM is much simpler than conventional methods.

Based on this study, we present some suggestions for future research. First, the detection performance of CUDM must be improved in more complex scenarios, such as those where the UAV is obscured at some point during its motion. In addition, the new detection principles proposed in this study may be considered for application to other objects.

## Figures and Tables

**Figure 1 sensors-22-09469-f001:**
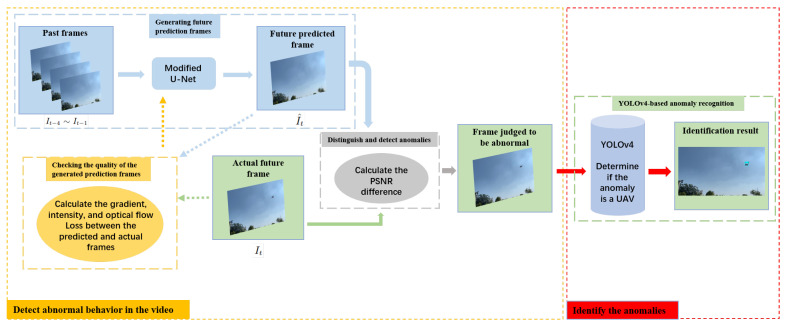
Framework and workflow of CUDM. The goal of the “detect abnormal behavior in the video” stage is to identify the frames containing abnormal objects, and the goal of the “identify the anomalies” stage is to determine whether the abnormal object is a UAV.

**Figure 2 sensors-22-09469-f002:**
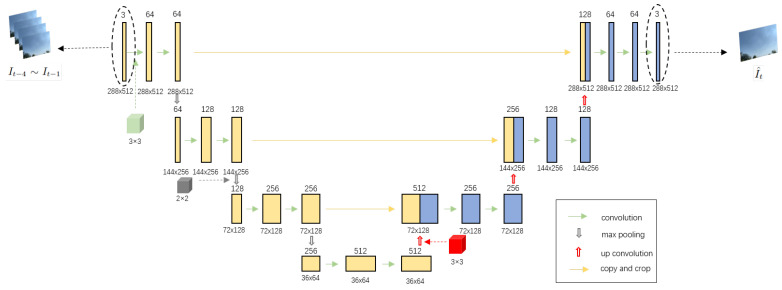
Structure of the modified U-Net network for generating predicted future frames.

**Figure 3 sensors-22-09469-f003:**
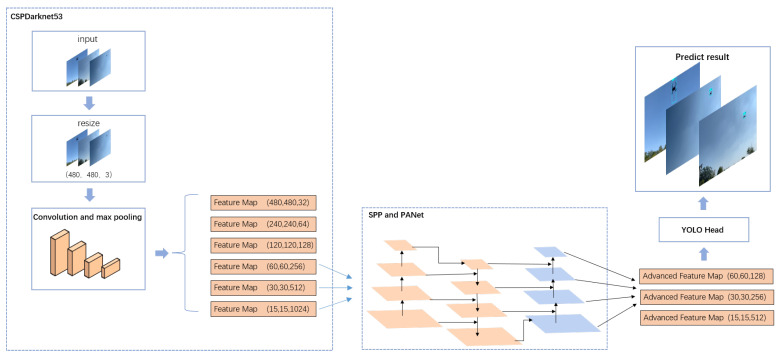
Structure and process of YOLOv4.

**Figure 4 sensors-22-09469-f004:**
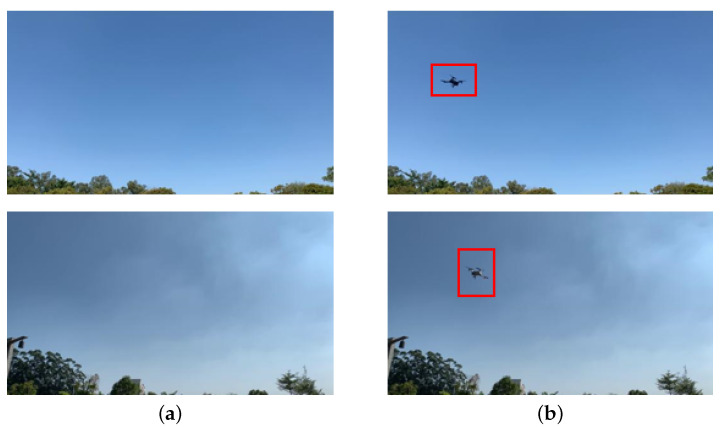
Part of the dataset in the stage of detecting abnormal behavior: (**a**) normal image, i.e., there is no object in the sky; (**b**) abnormal image, i.e., an object in the sky exists.

**Figure 5 sensors-22-09469-f005:**
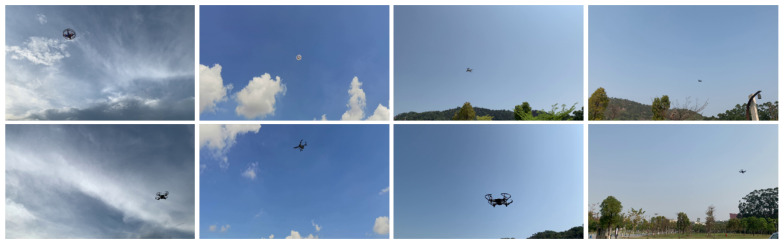
Part of the dataset for the anomaly identification stage.

**Figure 6 sensors-22-09469-f006:**
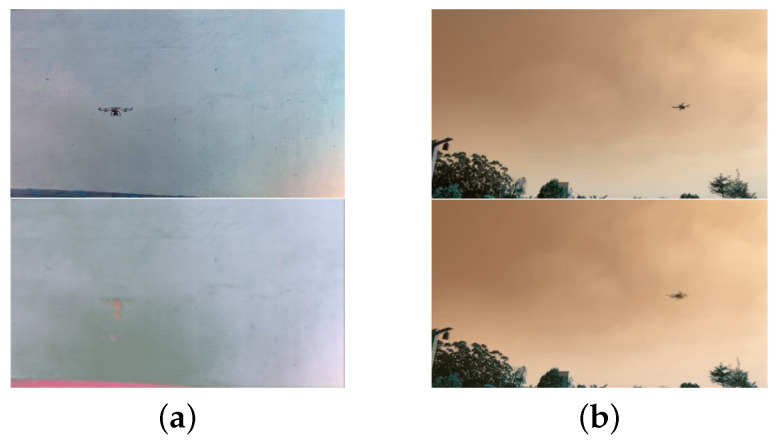
Two images generated by two sets of frames with different frame spacings (the actual frame is at the top while the generated frame is at the bottom): (**a**) frame generated by the discontinuous set, whose quality is significantly different from that of the actual frame; (**b**) frame generated by the continuous set, whose quality is significantly improved compared to that in (**a**).

**Figure 7 sensors-22-09469-f007:**
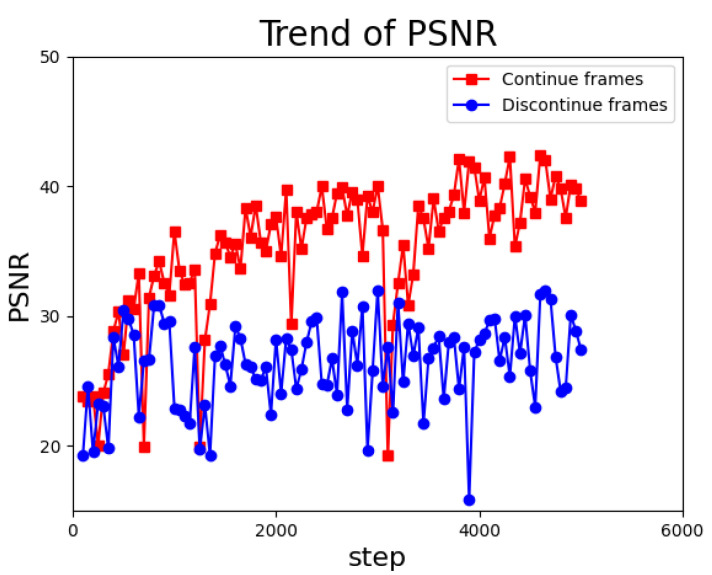
Graph of PSNR variation with higher peak PSNR and higher boost rate for consecutive frames.

**Figure 8 sensors-22-09469-f008:**
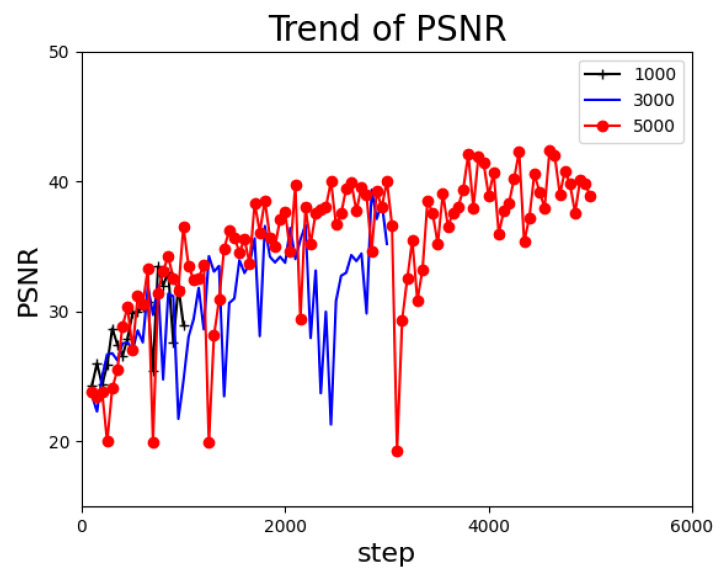
Graph showing the results of the PSNR trend when only the training duration is changed. The peak at 3000 steps is close to that at 5000 steps, and there is a significant drop in the peak value when the step is reduced to 1000.

**Figure 9 sensors-22-09469-f009:**
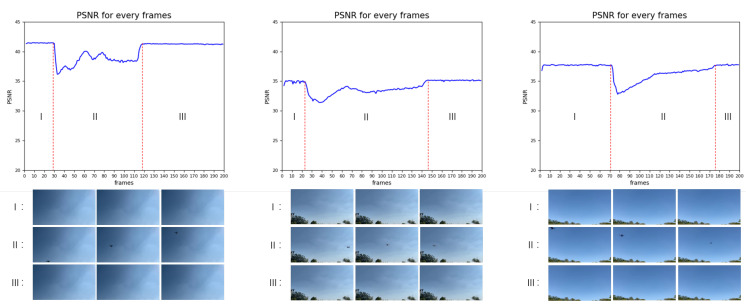
The two classification scenarios categorized in this study. Each frame has its own PSNR. The curves are divided into three stages: I and III are normal stages, whereas II is the abnormal stage where the UAV appears. In stage II, there is a significant drop in the PSNR curve, which can be used to determine the appearance of abnormalities. The bottom of the curve shows part of the frame of the phase corresponding to this curve.

**Table 1 sensors-22-09469-t001:** Overall performance comparison between CUDM and other models.

MODEL	mAP	Amount of Data	Average Recall	Average Precision	Average F1	Average AP
YOLOv3	79.22%	2.2k	82.63%	83.96%	0.83	79.22%
SSD	83.14%	2.2k	54.15%	94.17%	0.68	83.14%
FasterRCNN	64.63%	2.2k	64.69%	60.97%	0.62	64.63%
EfficienDet-d1	77.42%	2.2k	71.22%	85.27%	0.77	77.42%
CUDM	82.81%	1.5k	95.93%	71.47%	0.79	82.81%

**Table 2 sensors-22-09469-t002:** Comparison of detection results of four categories of UAVs between CUDM and other models.

Model	Category 1				Category 2				Category 3				Category 4			
	Recall	Precision	F1	AP	Recall	Precision	F1	AP	Recall	Precision	F1	AP	Recall	Precision	F1	AP
YOLOv3	86.96%	96.77%	0.92	87.04%	80.26%	77.22%	0.79	77.34%	72.16%	70.71%	0.71	61.43%	91.14%	91.14%	0.91	91.05%
SSD	65.22%	97.83%	0.78	90.85%	31.58%	92.31%	0.47	78.36%	46.39%	88.24%	0.61	67.32%	73.42%	98.31%	0.84	96.02%
FasterRCNN	75.36%	56.52%	0.65	72.33%	47.37%	56.25%	0.51	47.36%	75.26%	63.48%	0.69	74.03%	60.76%	67.61%	0.64	64.79%
EfficienDet-d1	76.81%	65.43%	0.71	76.70%	53.95%	85.42%	0.66	65.99%	68.04%	91.67%	0.78	73.16%	86.08%	98.55%	0.92	93.81%
CUDM	88.41%	24.40%	0.38	56.32%	97.37%	98.67%	0.98	97.37%	97.94%	77.87%	0.87	90.39%	100.00%	84.95%	0.92	87.15%

**Table 3 sensors-22-09469-t003:** Comparison of CUDM and other UAV inspection systems.

Model	Data Set Acquisition	Data Type	Variety of Experimental Scenarios	Low Experimental Costs	Real-Time Detection	UAV Detection Results
Phase-interferometric Doppler radar system [42]	Self-made	Radar signals	N	Y	Y	Y
Detection approach based on the machine learning [43]	Self-made	Radio Frequency	N	N	N	Y
Micro-UAV classification [44]	Self-made	Radio Frequency	N	N	N	Y
FastUAV-NET [45]	Self-made	UAV video	Y	Y	N	Y
Vision-based architecture [46]	Self-made	UAV image	N	N	N	Y
Anti-Drone System [11]	Self-made	Physical signals	Y	N	N	Y
CUDM	Self-made	UAV video	Y	Y	Y	Y

## Data Availability

Not applicable.

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
