# Peer review of "CUDM: A Combined UAV Detection Model Based on Video Abnormal Behavior"

_sensors, 2022, doi:10.3390/s22239469_

Round 1

Reviewer 1 Report

1.     The abstract section was recommended to be revised to add more details about the UAV detection model.

2.     Please explain what were the advantages and disadvantages for the deep learning related models for tacking the UAV detection methods.

3.      For Eq. (8) and (9), it seemed that a symbol was missed before 100%?

4.     The authors were recommended to add parameter setting procedure in the performance comparison section. For instance, how did authors set parameters for the YOLOx, etc.     

5.      For table 1, the authors were recommended to add more indicators to evaluate model performance.

6.     The authors were recommended to properly cite the following studies: [1] Utilizing UAV video data for in-depth analysis of drivers’ crash risk at interchange merging areas." Accident Analysis & Prevention 123 (2019): 159-169 [2]Ship detection from coastal surveillance videos via an ensemble Canny-Gaussian-morphology framework, Journal of Navigation, vol. 74, pp. 1252-1266, 2021.

Reviewer 2 Report

The authors propose a novel system for UAVs detection from videos and exploiting a deep learning-based approach.

The idea seems promising, it is also well presented, the architecture is clear, and its internal structure is too. However, we would like to suggest some improvements to improve the overall quality of the work.

In the introduction, the authors name numerous applications related to UAVs, but none of them refers to onboard captured images analysis. We suggest to expand that section, in order to provide a complete scenario for the reader, inserting the following references:

1) Eskandari, R.; Mahdianpari, M.; Mohammadimanesh, F.; Salehi, B.; Brisco, B.; Homayouni, S. Meta-analysis of Unmanned Aerial Vehicle (UAV) Imagery for Agro-environmental Monitoring Using Machine Learning and Statistical Models. Remote Sens. 202012, 3511. https://doi.org/10.3390/rs12213511

2) Avola, D.; Cinque, L.; Di Mambro, A.; Diko, A.; Fagioli, A.; Foresti, G.L.; Marini, M.R.; Mecca, A.; Pannone, D. Low-Altitude Aerial Video Surveillance via One-Class SVM Anomaly Detection from Textural Features in UAV Images. Information 202213, 2. https://doi.org/10.3390/info13010002

3) Zhao, J.; Zhang, X.; Yan, J.; Qiu, X.; Yao, X.; Tian, Y.; Zhu, Y.; Cao, W. A Wheat Spike Detection Method in UAV Images Based on Improved YOLOv5. Remote Sens. 202113, 3095. https://doi.org/10.3390/rs13163095

The presented structure of the architecture seems very effective; however, there is no ablation study that shows how some parameters are better than others (e.g., U-Net choice of layers type or sequence). We suggest introducing a small section in which the authors can specify why they make some choices instead of others.

An important section in the experimental phase is missing: comparing with similar systems. We noticed that the authors proposed a solution involving a self-made dataset, implying that the authors can only propose a comparison with standard well-known deep nets (e.g., FasterRCNN, YOLOv3,...). However, to improve the scientific soundness, it could be interesting to see some results obtained on public or well-known datasets for flying UAVs detection in a direct comparison with other systems in the state of the art that tried to provide a solution for that problem.

Concerning English, the manuscript is generally written in a good style; however, some improper forms and typos can be found extensively. We strongly suggest the authors to completely revise the paper to correct them.

Round 2

Reviewer 1 Report

My comments have been addressed.